# HAPEns: Hardware-Aware Post-Hoc Ensembling for Tabular Data

## Abstract

Ensembling is commonly used in machine learning on tabular data to boost predictive performance and robustness, but larger ensembles often lead to increased hardware demand. We introduce HAPEns, a post-hoc ensembling method that explicitly balances accuracy against hardware efficiency. Inspired by multi-objective and quality diversity optimization, HAPEns constructs a diverse set of ensembles along the Pareto front of predictive performance and resource usage. Experiments on 83 tabular classification datasets show that HAPEns significantly outperforms baselines, achieving superior accuracy–efficiency trade-offs. Ablation studies further reveal that memory usage is a particularly effective objective metric. Further, we show that even a greedy ensembling algorithm can be significantly improved in this task with a static multi-objective weighting scheme.

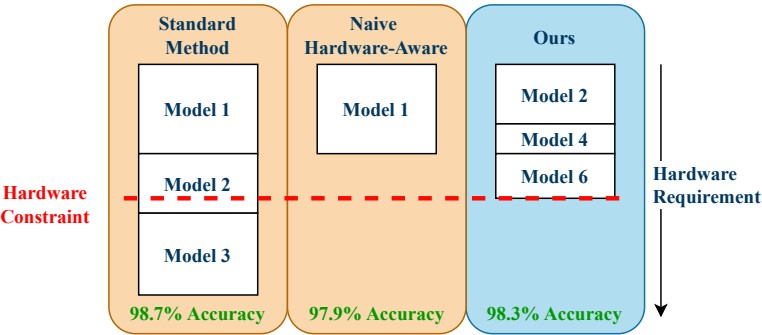

Figure 1: **Illustration of three ensemble selection strategies:** a standard method ignoring hardware constraints, a naive hardware-aware variant that sacrifices accuracy, and an advanced hardware-aware method that balances accuracy and efficiency. Box size reflects model resource usage; the red dashed line indicates the hardware resource constraint.

## 1 Introduction

Ensembling is a central technique in machine learning, used to improve predictive performance, stability, and robustness across a wide range of applications. From boosting and bagging in classical supervised learning to stacking in modern deep learning workflows, ensembles are frequently adopted to combine the strengths of diverse models. In many practical scenarios, models produced during training or exploratory analysis are later combined into ensembles in a post-hoc fashion to substantially improve performance (Erickson et al., 2025; Arango et al., 2025). This workflow has been further popularized by automated machine learning (AutoML) systems for tabular data (Purucker & Beel, 2023; He et al., 2021; Erickson et al., 2020), where greedy ensemble selection (GES) by Caruana et al. (2004) has emerged as a widely used method to automatically build strong ensembles from model libraries.

While post-hoc ensembling generally improves predictive performance, larger ensembles lead to increased hardware demands at inference time. Each additional model increases prediction latency and resource consumption, inducing higher costs. While this matters greatly in production settings, it is ignored by standard post-hoc ensembling methods. As machine learning is increasingly deployed in environments with tight resource constraints, the gap between high predictive accuracy and hardware feasibility has become more pronounced.

We address this challenge by introducing HAPEns, a post-hoc ensembling method that explicitly balances predictive performance against hardware costs. It improves on existing baselines by constructing Pareto fronts of ensembles that more effectively balance competing objectives. Thus, practitioners can select better models that satisfy both performance and deployment requirements under their specific hardware constraints. Drawing inspiration from multi-objective optimization (Gunantara, 2018) and quality diversity optimization (Pugh et al., 2016), HAPEns maintains a diverse population of ensembles that vary in both hardware cost and predictive behavior, while optimizing for predictive performance. The result is a set of candidate ensembles that offer distinct trade-offs between both objectives.

To evaluate HAPEns, we performed experiments on 83 tabular classification datasets of varying size and complexity. We compare ensembles constructed by our method to those selected by baselines like GES and a novel multi-objective baseline. Our findings reveal that optimizing for memory footprint is a particularly effective metric for deployment cost and that our method significantly outperforms competitors in balancing hardware costs and predictive performance.

**Our Contributions.** In this work, we: (i) Propose a novel post-hoc ensembling algorithm that explicitly incorporates hardware cost into the selection process; (ii) Demonstrate through extensive benchmarking that our method achieves superior accuracy–cost trade-offs compared to existing baselines; (iii) Show that memory-awareness yields substantial gains even in inference-time efficiency; (iv) Ensure reproducibility by open-sourcing all code[1], results, and integration with popular ensembling frameworks.

## 2 Related Work

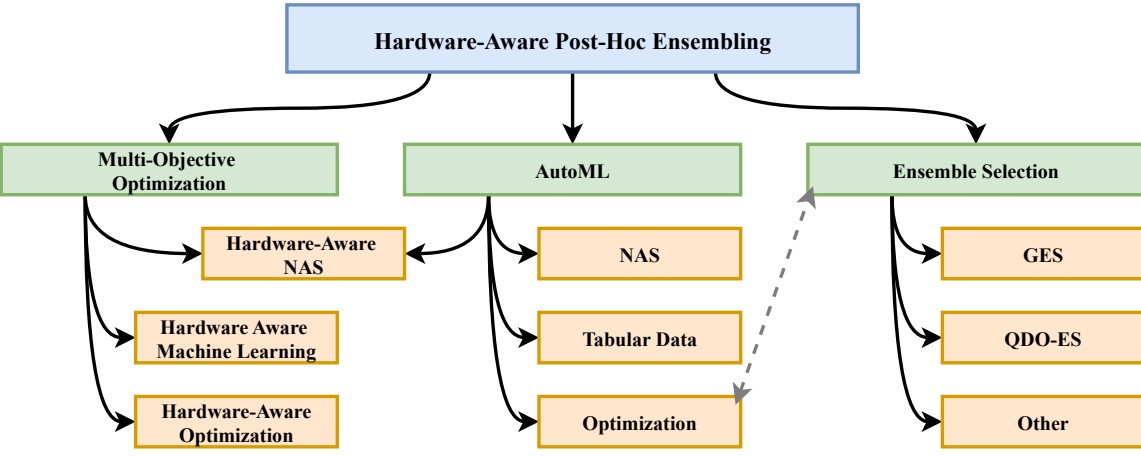

Figure 2: An overview of the main research areas for this paper showing the intricate interdisciplinary connections.

Ensembling—combining multiple pre-trained models—is an effective approach to improve predictive performance and robustness. Common strategies include bagging, stacking, and ensemble selection (ES). Bagging and stacking are typically integrated into the training process, whereas ES can be applied post hoc, that is, after model training has completed. ES might then also be referred to as post hoc ensembling.

---

[1]All code used for this publication is available at: https://anonymous.4open.science/r/C07F

ES as introduced by Caruana et al. (2004) is a forward selection algorithm that greedily constructs an ensemble by iteratively adding the model that improves the predictive performance of the ensemble the most. The resulting ensemble is defined by a weight vector derived from this superset of selected models. In this work, we adopt a broader interpretation of ensemble selection: Any algorithm that produces such a weight vector from a pool of trained models qualifies as ES. To distinguish the classical algorithm of Caruana et al., we refer to it as greedy ensemble selection (GES).

Post-hoc ensembling is a widely adopted component in automated machine learning (AutoML) systems, particularly for tabular data (Erickson et al., 2020; Feurer et al., 2015; Purucker & Beel, 2023). It enables the reuse of models generated during training without retraining, making it a computationally attractive final optimization step. Although ensemble selection can theoretically be used during training, we reserve the term ES for its post-hoc usage in this work. The term blending also appears in this context, but it specifically refers to ensemble selection applied to a holdout validation set distinct from the training data.

Recent years have seen the integration of multi-objective optimization (MOO) into various stages of the machine learning and AutoML pipelines, including neural architecture search (NAS) (Benmeziane et al., 2021b;a). These methods optimize for trade-offs such as accuracy versus latency, energy consumption, or memory usage. However, the use of MOO techniques in post-hoc ensemble selection remains largely unexplored. Shen et al. (2022) introduced DivBO, a diversity-aware Bayesian optimization framework that incorporates ensemble selection during candidate evaluation to promote both accuracy and diversity. Although their approach targets the model search stage rather than post-hoc optimization, it highlights the potential of multi-objective formulations to improve ensemble composition. Nevertheless, to the best of our knowledge, no prior work systematically investigates the construction of Pareto-optimal ensembles that explicitly account for hardware constraints such as inference time or memory usage.

Modern implementations of GES—still the de facto standard in AutoML frameworks such as Auto-sklearn (Feurer et al., 2022) and AutoGluon (Erickson et al., 2020)—typically optimize only for predictive performance and remain agnostic to deployment cost. Consequently, they may produce ensembles that are unnecessarily large or infeasible for deployment due to hardware requirements.

Our work addresses this identified gap by introducing a hardware-aware approach to ES, explicitly targeting the trade-off between accuracy and resource usage. QDO-ES as developed by Purucker et al. (Purucker et al., 2023) inspired HAPEns and the inclusion hardware metrics during ensemble construction. In doing so, we extend the utility of ES beyond predictive performance to deployment and real-world use.

## 3 Method

One of the last steps in the ML pipeline is model generation, where human experts or AutoML systems explore and evaluate various configurations. This process yields a set of candidate models, typically followed by the selection of the single best model for deployment. Post-hoc ensembling instead aims to improve the quality of the prediction by combining multiple candidates from this set.

Let $\mathcal{M} = \{M_1, \ldots, M_p\}$ be the library of models and let $c_j$ be the number of times $M_j$ is selected out of a total of $T$ picks (with repetition). Define the weight vector $\mathbf{w}$:

$$\mathbf{w} = (w_1, \ldots, w_p)^\top = \frac{1}{T}(c_1, \ldots, c_p)^\top, \quad w_j = \frac{c_j}{T}, \quad \sum_j w_j = 1. \tag{1}$$

The ensemble predictor for input $x$ is $f_{\text{ens}}(x) = \sum_{j=1}^p w_j \, f_j(x)$. This formulation applies broadly: for regression, each $f_j(x)$ is a scalar prediction; for probabilistic classification, $f_j(x)$ is a vector of class probabilities, and $f_{\text{ens}}(x)$ is the averaged probability vector.

Although the ensemble predictor is ultimately defined by a weight vector, there are multiple ways to construct it. A common method is GES, which uses a forward selection strategy to iteratively build the ensemble by greedily adding models that improve performance the most. In contrast, our work explores a population-based approach.

We begin by sampling an initial population of ensembles across a two-dimensional behavior space (e.g., memory footprint vs. average loss correlation). Each ensemble is evaluated and stored in a niche corresponding to its behavior and hardware costs. New ensembles are generated by selecting suitable parents from these niches and applying crossover and mutation (see Figure 3). This process repeats until convergence or a time/iteration limit is reached, allowing us to explore a wide range of model combinations and discover Pareto-optimal trade-offs between prediction quality and deployment cost. What follows now are detailed definitions of these concepts, similarly outlined by Purucker et al. (2023).

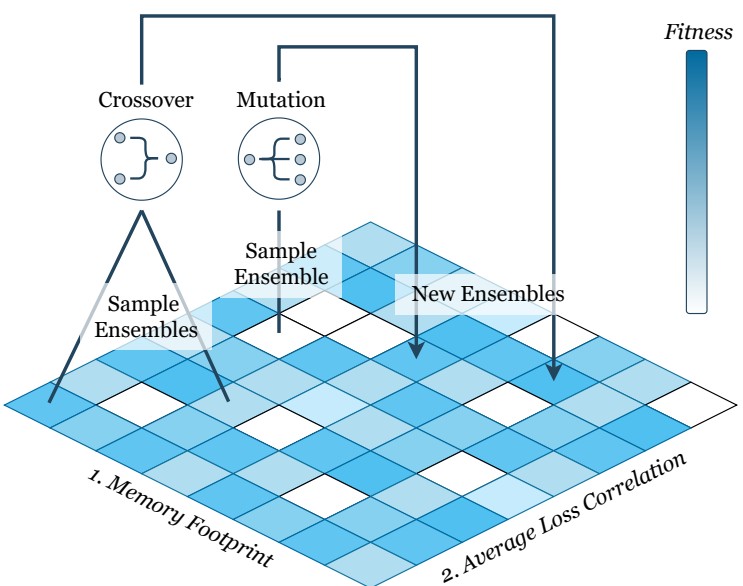

Figure 3: **Illustration of the HAPEns search process.** Ensembles are sampled from bins over memory footprint and average loss correlation, then evolved via crossover and mutation to explore the behavior space.

**Behavior Space.**  Each ensemble $E$ is assigned a two-dimensional descriptor $b(E) = (\text{ALC}, \text{HW})$, where ALC is the mean Pearson correlation among the loss vectors of its constituent models and HW is a hardware metric aggregated over those models. Following prior work (Purucker et al., 2023), we divide this 2D space into a 7 by 7 grid using a sliding bounding archive (Fontaine et al., 2019), creating 49 bins (niches). The algorithm allows ensembles to compete only within the same niche. This ensures that different regions of the behavior space can retain their best solutions. Therefore, a diverse population across two objectives is maintained while optimizing predictive performance. Here we found that memory as a hardware metric produces ensembles which are best at trading off predictive performance and hardware cost.

**Fitness.**  Each ensemble $E$ is scored by a scalar loss $L(E)$ on cross-validation data. The behavior space is partitioned into fixed niches or bins, and each niche retains the lowest loss ensemble observed.

**Sampling.**  The parents are selected from the archive using a combined dynamic strategy that balances exploration and exploitation. The method alternates between deterministic selection of the best solution and stochastic selection of random solutions, with the selection probability dynamically adjusted every ten iterations based on which approach yields better results. Deterministic and tournament-based selection methods are also available as alternatives.

**Crossover.**  Two parent weight vectors are recombined by a two-point crossover limited to nonzero entries or by averaging their counts and rounding up to maintain valid repetition counts.

**Mutation.**  The count of a single model in the repetition vector is increased at random, with rejection sampling used to avoid duplicates (up to 50 retries).

## 4 Experimental Setup

The main objective of this paper is to compare the proposed method and the baselines on how well they can balance predictive performance and hardware costs. In this context, only Pareto optimal ensembles are relevant. In addition, there is no best ensemble because choosing the right trade-off depends on the real world scenario. Therefore, our main focus lies with the Pareto fronts of ensembles generated by each method.

Our proposed method uses memory usage as its hardware-aware behavior metric. We compare it with four baselines:

- **Single-Best**: A naive baseline that selects the single model with the highest validation performance. Including Single-Best highlights the performance gains achieved through ensembling.

- **GES\***: Our implementation of greedy ensemble selection (GES) enhanced to return the entire sequence of ensembles generated during its run. GES* therefore represents the best-case performance of the original widely used GES, providing a strong reference point for assessing improvements.

- **Multi-GES**: Our implementation of novel multi-objective extensions of GES to enable the algorithm to balance predictive performance and inference time using a static weighting scheme; see Appendix A.1 for details on our implementation. Multi-GES reflects a straightforward approach to introducing hardware awareness into ensemble selection and allows us to assess the benefits of our more flexible formulation.

- **QDO-ES**: The quality-diversity optimization ensemble selection method(Purucker et al., 2023), which optimizes for performance and behavioral diversity but is not hardware aware. This baseline isolates the effect of hardware awareness by comparing against a method that can already generate various Pareto-optimal ensembles without considering resource costs.

To assess the quality of the generated Pareto fronts, we rely on two standard multi-objective indicators: inverted generational distance plus (IGD+) (Ishibuchi et al., 2015) and hypervolume (HV) (Zitzler & Thiele, 1999). IGD+ quantifies how well a set of solutions approximates a reference front, which in our case is constructed from the Pareto optimal solutions of all the methods under comparison. HV measures the portion of the objective space dominated by a set of solutions (see B for details). The set of solutions here is the set of ensembles constructed by one method for a given task and seed. Both HV and IGD+ are widely used in multi-objective evaluation, and for our experiments we employ the `pygmo` (Biscani & Izzo, 2020) implementation. We focus primarily on HV in the main analysis because we do not have a true Pareto front for IGD+, and both metrics lead to the same conclusions.

The normalization of the ROC AUC and the hardware metrics was done per seed and task and over all methods tested in this paper. This makes the results comparable across experiments even after selecting specific methods per experiment. To ensure a comprehensive and reproducible evaluation, we organize our experiments into three groups shown in Table 1.

**Datasets.** We conducted our experiments using TabRepo(Salinas & Erickson, 2023), which provides precomputed model predictions for 1,530 models in 211 tabular regression and classification datasets. This allowed for a large-scale, reproducible simulation of the post-hoc ensemble selection process.

We experimented with the `D244_F3_C1530_100` context of TabRepo, utilizing 10 different seed settings. Within this framework, a total of 100 different datasets were available, from which regression sets were excluded. The characteristics of the remaining 83 classification datasets are shown in Figure 4, which reveals a wide variety of class, sample, and feature counts. Consequently, the results provide a comprehensive overview of the performance of the tested methods across various use cases.

The available models in TabRepo are plotted in Figure 5 with their respective inference times for different tasks. The violin plot shows the variety of models in TabRepo, from cheap boosting and linear regression algorithms to the more computationally expensive transformers. Thus, the base models in our experiments comprise a diverse set that the evaluated ensemble construction methods can utilize.

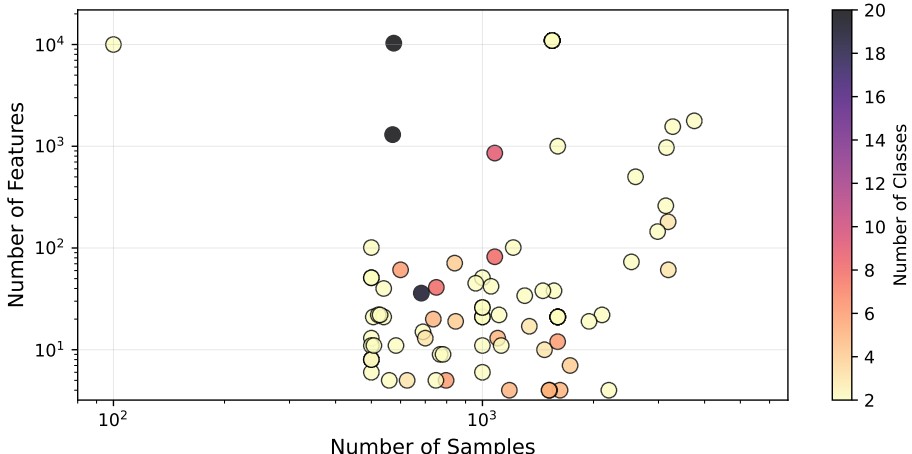

Figure 4: Scatter plot of datasets over their number of features (y), number of samples (x), and the number of classes (color).

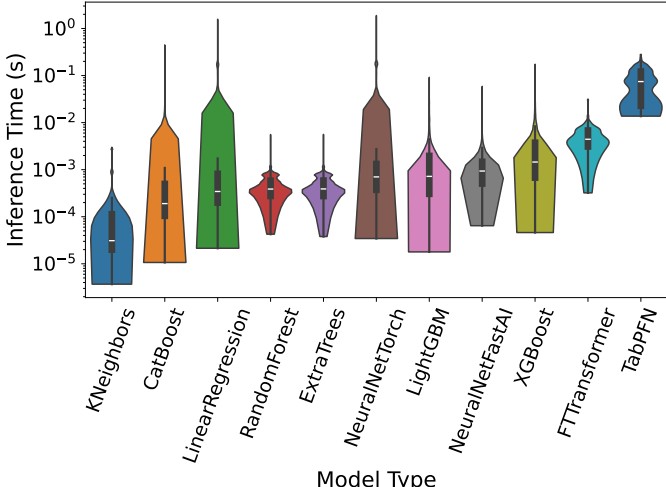

Figure 5: Comparison of TabRepos model types and their corresponding inference times for varying tasks. KNeighbours and linear regression are expectedly on the lower end of the spectrum, while transformers have increased cost due to their complexity.

Table 1: Overview of experiments

| Group | ID | Description |
|-------|-----|-------------|
| Main Results | EXP1 | Evaluation of HAPEns and baseline methods using the HV indicator, calculated from the ROC AUC of the test (inverted to a loss) and the averaged normalized hardware metrics: inference time, memory usage, disk space usage. |
| Details | EXP2 | Comparison of HV and IGD+ values to assess robustness between indicators. |
| | EXP3 | HV evaluation with respect to each individual hardware metric (inference time, memory, disk space). |
| | EXP4 | Examining the differences in the construction of the ensemble between HAPEns and the baseline methods. |
| Ablation | EXP5 | Analysis of alternative hardware metrics in the behavior space of HAPEns. |
| | EXP6 | Sensitivity analysis of different weightings in the Multi-GES method. |

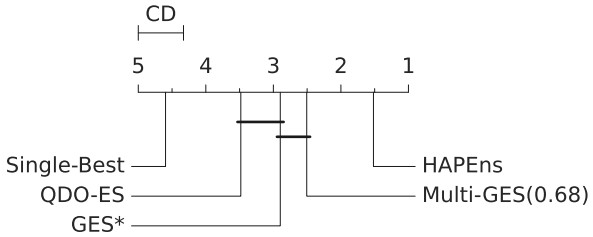

Figure 6: HAPEns significantly outperforms the baselines on HV. Single-Best is significantly outperformed by all other methods.

Figure 7: HAPEns significantly outperforms the baselines on IGD+. Single-Best is significantly outperformed by all other methods.

## 5    Results

We first present the main results, followed by detailed analyses, and finally ablation studies. The central focus is on the ability of each method to balance two objectives: predictive performance and hardware cost. In particular, identifying a single strong ensemble may be less effective than discovering several competitive ensembles that trade off these objectives differently. In general, HAPEnsconsistently outperforms baselines in both Main Results (EXP1) and Details (EXP2, EXP3, EXP4), demonstrating its superior ability to produce competitive ensembles while incorporating hardware awareness.

**Main Results (EXP1)**

EXP1    Figure 6 shows a critical difference (CD) diagram (Demšar, 2006; Herbold, 2020) summarizing the average ranks of the methods evaluated based on their HV values. The HV was calculated from the inverted ROC AUC on the test data and the averaged normalized hardware metrics (inference time, memory, and disk usage)—collectively referred to as the *hardware score*. Therefore, this figure provides an overview for all the datasets, model configurations, and hardware metrics we explored in our tests. To simplify the presentation and highlight the overall trade-off between predictive performance and hardware costs, we aggregate the three hardware measures into a single score. This avoids overemphasizing any single metric, while keeping the focus on the general notion of hardware efficiency.

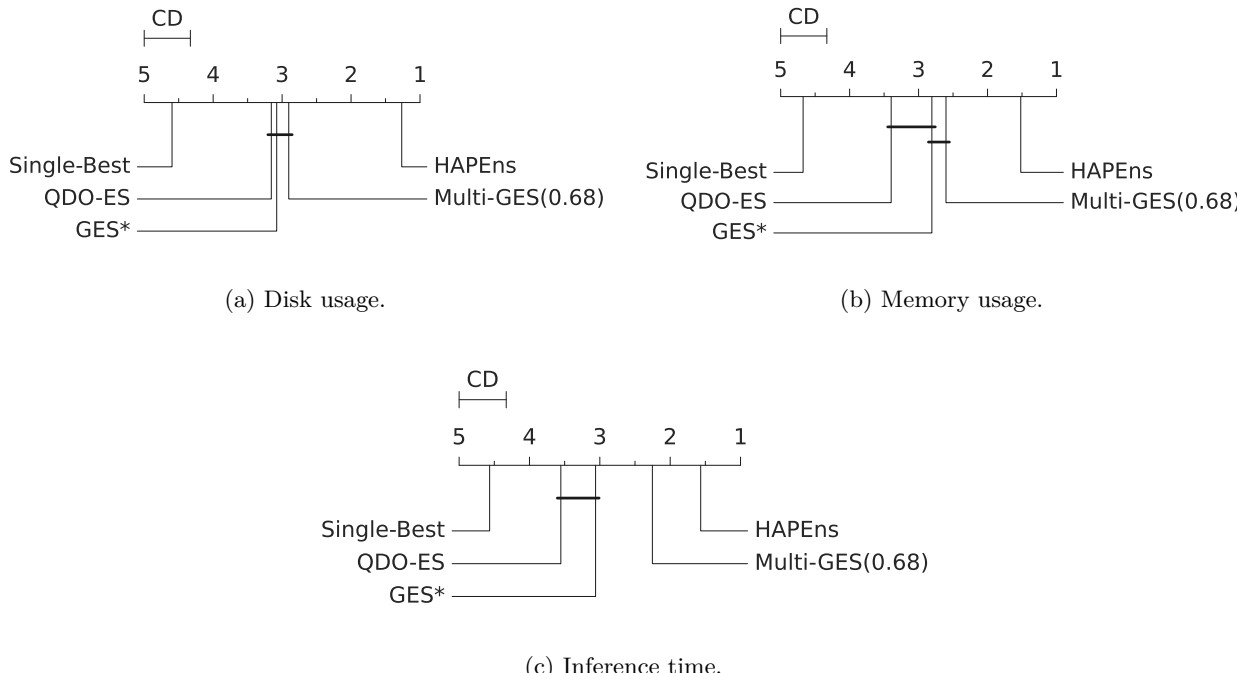

(a) Disk usage.

(b) Memory usage.

(c) Inference time.

Figure 8: Critical difference plots for the hypervolume across different hardware-aware objectives.

In the CD plot, methods connected by a horizontal bar are statistically indistinguishable according to the Nemenyi post-hoc test. HAPEns shows significantly superior performance to the baselines, which makes it the best method to balance the trade-off between predictive performance and hardware costs. Between the baselines, we do not see significant differences except for the single-best method, which simply picks the best model configuration based on its ROC AUC. A single-best model is not well suited for this setting because it cannot capture diverse trade-offs between predictive performance and different hardware costs, which multiple ensembles can exploit more effectively. We see slight improvements in GES* over QDO-ES, which can be attributed to the modification of GES to return all intermediate ensembles, which generally leads to a higher number of ensembles produced (see Figure 10). This improvement over the standard procedure of returning the final ensemble gives GES a strong edge here. Multi-GES performs slightly higher, but insignificantly so, by constructing ensembles with reduced hardware costs while keeping their predictive performance comparable to GES*. A discussion on GES*'s overfitting problem and the corresponding cost-to-performance trade-off follows in the Multi-GES ablation part of this section.

**Details (`EXP2`, `EXP3`, `EXP4`)**

`EXP2`   In Figure 7, the IGD+ results are generally consistent with the HV findings. The main difference is the stronger relative performance of Multi-GES, which now significantly outperforms GES and comes close to matching HAPEns, to the point that HAPEns's superiority is no longer statistically significant. This effect arises because Multi-GES constructs more efficient ensembles, while QDO-ES primarily improves predictive performance (Figure 9) but at the cost of building more expensive ensembles on average. Since IGD+ evaluates solutions with respect to a reference front, Multi-GES benefits disproportionately: a larger share of its efficient solutions lies on the reference Pareto front, reducing the relative advantage of HAPEnscompared to dominated HV. For this reason, we focus on HV in the remainder of the paper, while noting that Multi-GES is particularly strong at exploring the low-cost end of the Pareto front.

`EXP3`   Looking at the HV results for the individual hardware metrics in Figure 8, we see in more detail what was already evident in the main results: HAPEns performs strongly across all metrics. The method demonstrates robustness to different hardware considerations, even when the behavior space is defined solely

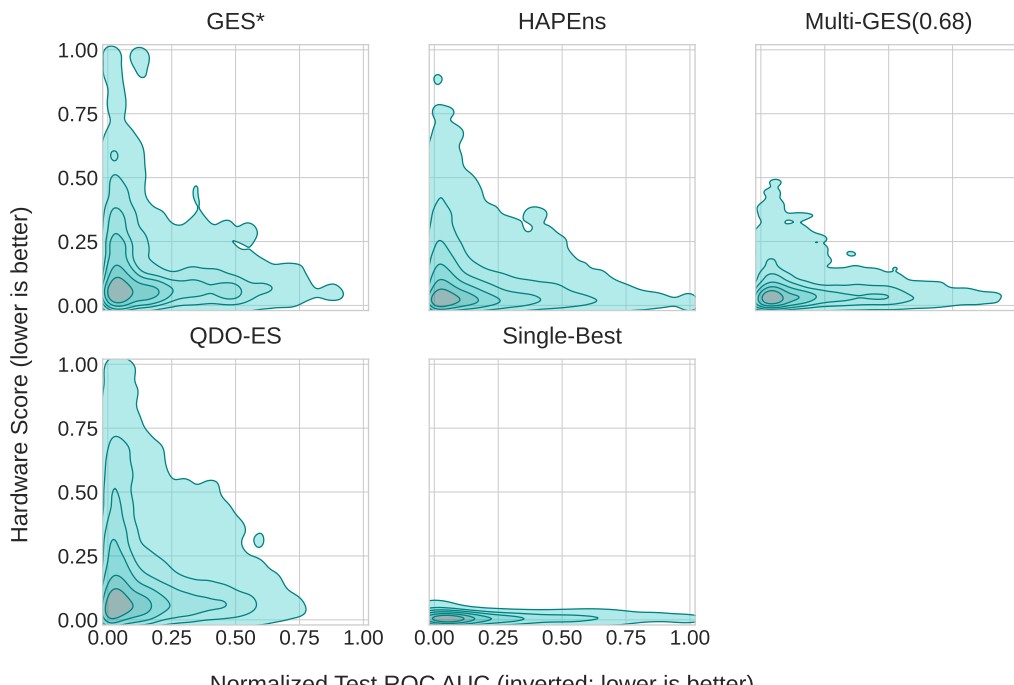

Figure 9: Comparison of constructed ensembles when including hardware metrics in the ensembling process. The baselines and the hardware-aware methods in the density plots, produce a clear trend, where the ensembles of the latter methods are more condensed toward the x-axis.

by memory usage. Notably, Multi-GES shows a significant improvement over the other baselines when optimizing for inference time. This highlights its specialization toward the specific hardware metric it uses during ensemble construction, but also suggests that inference time may generalize less effectively across metrics compared to memory. Since our experiments abstract away from specific hardware configurations, these findings should be viewed as preliminary. Overall, these results point to an interesting direction for future research that investigates hardware-aware behavior more directly under diverse configurations and cost measures.

**EXP4**  Figure 9 shows a density plot of the ensembles constructed by the different methods. Compared to Single-Best, all ensemble methods increase hardware costs but also yield clear gains in predictive performance. Multi-GES reduces hardware costs relative to GES*, confirming its intended effect. QDO-ES and HAPEns produce similar overall trends, but the ensembles of HAPEns are more concentrated along the x-axis, indicating lower resource usage. These observations clarify and reinforce the improvements of HAPEns over QDO-ES in terms of hardware efficiency, and likewise of Multi-GES over GES*. Overall, the inclusion of hardware metrics in the ensemble construction process achieves the desired shift toward more efficient ensembles.

Figures 10 and 11 provide additional insight into the behavior of the tested methods. GES* produces 10–15 more ensembles on average than HAPEns, yet fewer of them lie on the Pareto front, indicating that many of its ensembles are not useful in this context. QDO-ES and HAPEns both generate a high ratio of unique ensembles, illustrating the effectiveness of the behavior space in promoting diversity. By contrast, Multi-GES produces fewer ensembles overall and fewer unique ensembles than GES*, which aligns with the increased difficulty of adding models once hardware costs are incorporated into the selection process.

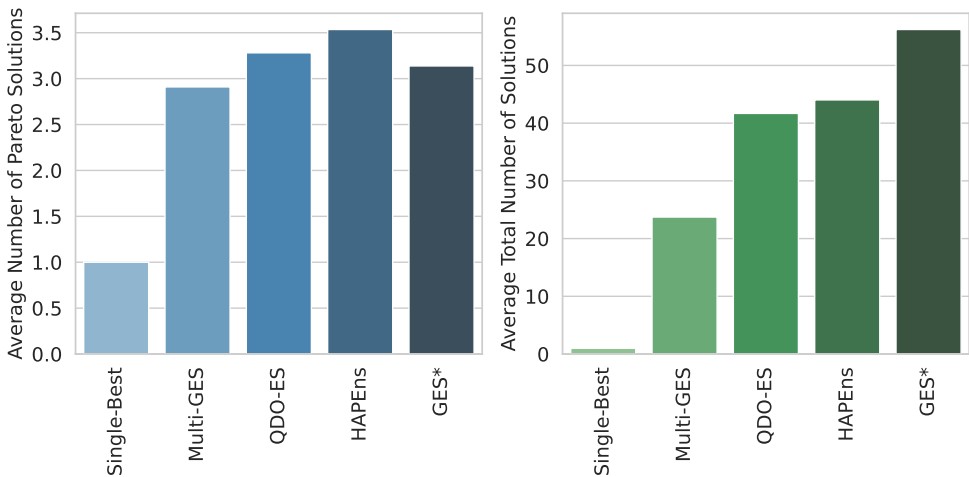

Figure 10: The average number of Pareto ensembles created vs the total number of ensembles created per method on average. The averages are per seed and task to reflect real-world yields of these methods.

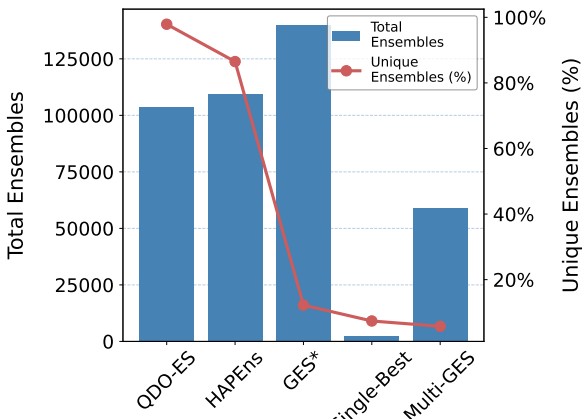

Figure 11: Overall number of ensembles generated per method and how many of these were unique when comparing the specific weight vectors.

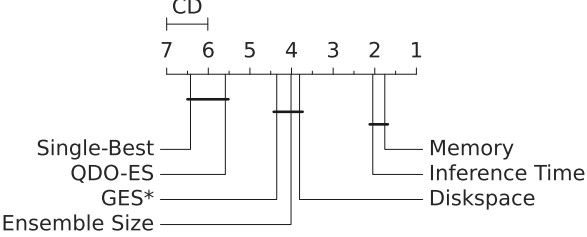

Figure 12: Comparison of different hardware metrics used for HAPEns. Memory and inference time perform strongest, but ensemble size is still notable as a proxy hardware metric, which does not need additional measurements.

**Ablation (`EXP5`, `EXP6`)**

**`EXP5`** We further evaluated HAPEns with four different hardware metrics: inference time, memory usage, disk usage, and ensemble size. The last serves only as a proxy hardware metric, yet Figure 12 shows that it still provides a competitive signal to balance the trade-off, without requiring additional measurements. Among the true hardware metrics, memory usage and inference time consistently lead to the strongest results, with memory showing a slight edge. These findings highlight that, while the size of the ensemble can act as a lightweight approximation, the use of actual hardware metrics yields the most reliable improvements.

**`EXP6`** In Figures 13 and 14 we investigate the effect of different static weightings in Multi-GES. By gradually increasing the weight on the inference time, the constructed ensembles shift from high-performing but more expensive configurations toward ensembles with lower hardware costs. This transition is clearly visible in the density plots, where the mass of ensembles moves closer to the origin of the objective space as the

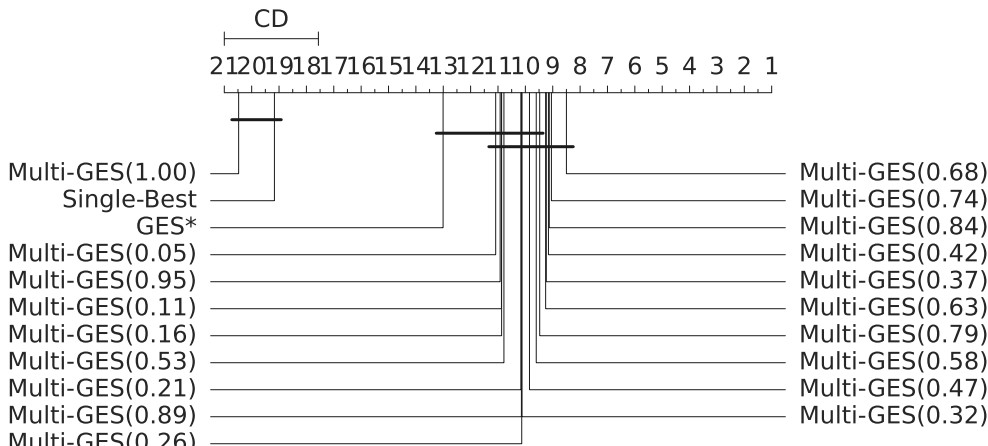

Figure 13: Comparison of static weights for Multi-GES highlighting the trade-off between predictive performance and hardware costs.

emphasis on inference time increases. The trade-off between predictive performance and efficiency becomes apparent: a stronger emphasis on time reduces costs but slightly lowers predictive accuracy, while a weaker emphasis maintains accuracy at the expense of efficiency. In Figure 13 we see a sweet spot, where excessively high or low time weights yield sub-par performance relative to intermediate weightings. For comparison in the main results, we chose the best performing weight: 0.68. These results confirm that Multi-GES allows practitioners to explicitly control the desired balance between performance and hardware costs through a weighting mechanism, highlighting its flexibility for different deployment scenarios.

## 6 Conclusion

This work introduced HAPEns, a hardware-aware post hoc ensemble selection method that explicitly balances predictive performance and deployment efficiency. By integrating hardware metrics into the ensemble construction process, HAPEns extends traditional greedy ensemble selection into a multi-objective optimization framework that explores the Pareto front of accuracy and resource usage. Across 83 diverse tabular classification datasets, HAPEns consistently outperforms existing baselines, achieving superior accuracy–efficiency trade-offs and demonstrating robustness across different hardware cost metrics.

Ablation studies reveal that memory usage is a particularly effective objective, providing a stable optimization signal and leading to ensembles that generalize well across cost measures. Additionally, our experiments show that even simple greedy methods like GES can benefit substantially from static multi-objective weighting, emphasizing the broad potential of hardware-aware ensemble construction.

Future work may explore dynamic weighting schemes, task-specific hardware profiling, and integration into end-to-end AutoML pipelines to further improve deployment efficiency without compromising predictive performance. Ultimately, HAPEns bridges the gap between high predictive accuracy and real-world hardware constraints, offering a practical and scalable step toward more sustainable machine learning systems.

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

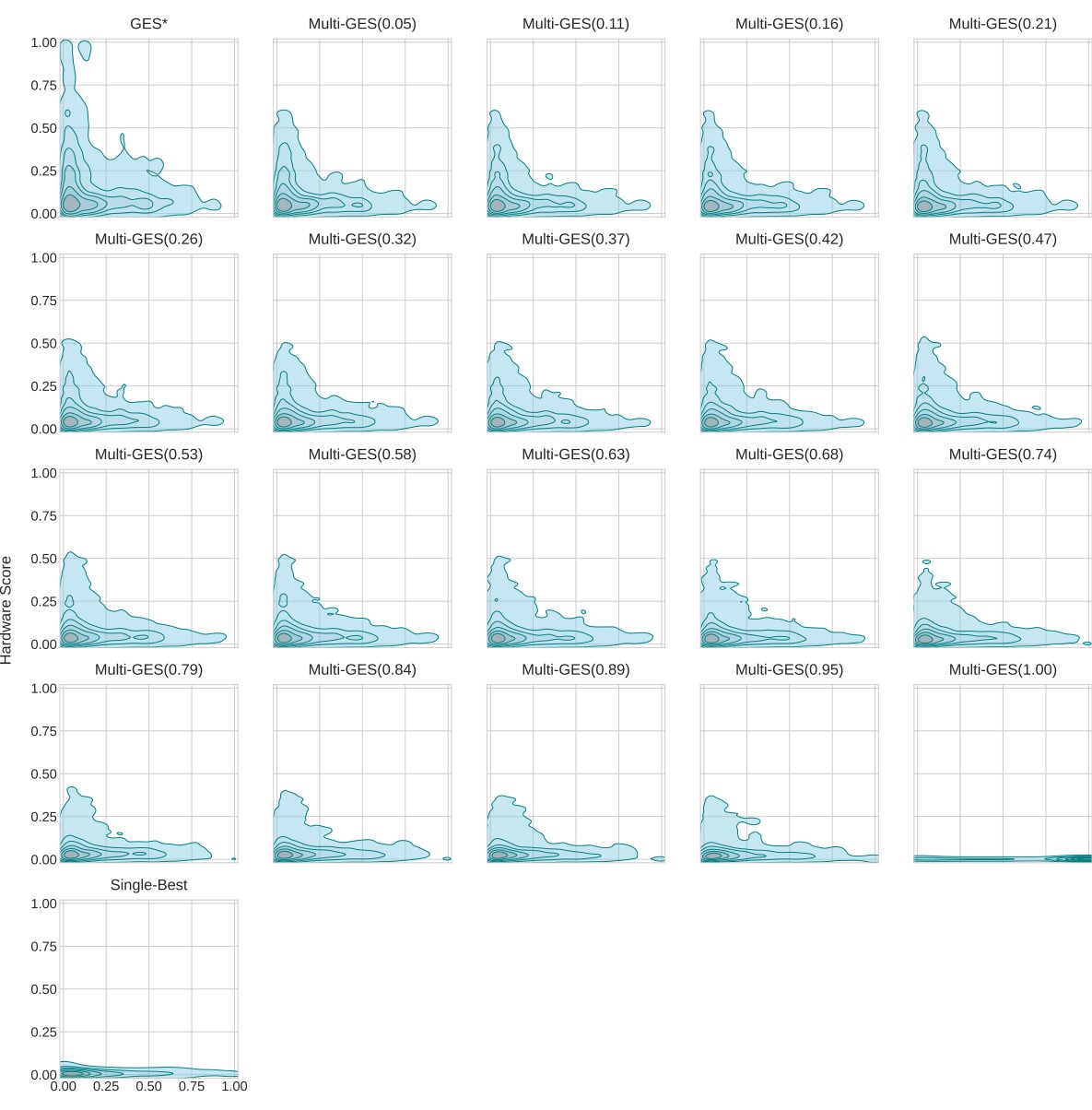

Figure 14: Density of ensembles produced by Multi-GES transitions to less expensive ensembles when increasing the time weight.

Hadjer Benmeziane, Kaoutar El Maghraoui, Hamza Ouarnoughi, Smail Niar, Martin Wistuba, and Naigang Wang. A Comprehensive Survey on Hardware-Aware Neural Architecture Search, January 2021b. URL http://arxiv.org/abs/2101.09336. arXiv:2101.09336 [cs].

Francesco Biscani and Dario Izzo. A parallel global multiobjective framework for optimization: pagmo. *Journal of Open Source Software*, 5(53):2338, 2020. doi: 10.21105/joss.02338. URL https://doi.org/10.21105/joss.02338. Publisher: The Open Journal.

Rich Caruana, Alexandru Niculescu-Mizil, Geoff Crew, and Alex Ksikes. Ensemble selection from libraries of models. In *Proceedings of the twenty-first international conference on Machine learning*, ICML '04, pp. 18, New York, NY, USA, July 2004. Association for Computing Machinery. ISBN 978-1-58113-838-2. doi: 10.1145/1015330.1015432. URL https://doi.org/10.1145/1015330.1015432.

Janez Demšar. Statistical comparisons of classifiers over multiple data sets. *The Journal of Machine learning research*, 7:1–30, 2006. URL https://www.jmlr.org/papers/volume7/demsar06a/demsar06a.pdf. Publisher: JMLR. org.

Nick Erickson, Jonas Mueller, Alexander Shirkov, Hang Zhang, Pedro Larroy, Mu Li, and Alexander Smola. AutoGluon-Tabular: Robust and Accurate AutoML for Structured Data, March 2020. URL http://arxiv.org/abs/2003.06505. arXiv:2003.06505 [cs, stat].

Nick Erickson, Lennart Purucker, Andrej Tschalzev, David Holzmüller, Prateek Mutalik Desai, David Salinas, and Frank Hutter. TabArena: A Living Benchmark for Machine Learning on Tabular Data, June 2025. URL http://arxiv.org/abs/2506.16791. arXiv:2506.16791 [cs].

Matthias Feurer, Aaron Klein, Katharina Eggensperger, Jost Springenberg, Manuel Blum, and Frank Hutter. Efficient and Robust Automated Machine Learning. In *Advances in Neural Information Processing Systems*, volume 28. Curran Associates, Inc., 2015. URL https://proceedings.neurips.cc/paper_files/paper/2015/hash/11d0e6287202fced83f79975ec59a3a6-Abstract.html.

Matthias Feurer, Katharina Eggensperger, Stefan Falkner, Marius Lindauer, and Frank Hutter. Auto-sklearn 2.0: Hands-free automl via meta-learning. *Journal of Machine Learning Research*, 23(261):1–61, 2022. URL https://www.jmlr.org/papers/v23/21-0992.html.

Matthew C. Fontaine, Scott Lee, L. B. Soros, Fernando De Mesentier Silva, Julian Togelius, and Amy K. Hoover. Mapping hearthstone deck spaces through MAP-elites with sliding boundaries. In *Proceedings of the Genetic and Evolutionary Computation Conference*, GECCO '19, pp. 161–169, New York, NY, USA, July 2019. Association for Computing Machinery. ISBN 978-1-4503-6111-8. doi: 10.1145/3321707.3321794. URL https://doi.org/10.1145/3321707.3321794.

Nyoman Gunantara. A review of multi-objective optimization: Methods and its applications. *Cogent Engineering*, 5(1):1502242, January 2018. ISSN 2331-1916. doi: 10.1080/23311916.2018.1502242. URL https://www.tandfonline.com/doi/full/10.1080/23311916.2018.1502242.

Xin He, Kaiyong Zhao, and Xiaowen Chu. AutoML: A survey of the state-of-the-art. *Knowledge-Based Systems*, 212:106622, January 2021. ISSN 0950-7051. doi: 10.1016/j.knosys.2020.106622. URL https://www.sciencedirect.com/science/article/pii/S0950705120307516.

Steffen Herbold. Autorank: A Python package for automated ranking of classifiers. *Journal of Open Source Software*, 5(48):2173, April 2020. ISSN 2475-9066. doi: 10.21105/joss.02173. URL https://joss.theoj.org/papers/10.21105/joss.02173.

Hisao Ishibuchi, Hiroyuki Masuda, Yuki Tanigaki, and Yusuke Nojima. Modified Distance Calculation in Generational Distance and Inverted Generational Distance. In António Gaspar-Cunha, Carlos Henggeler Antunes, and Carlos Coello Coello (eds.), *Evolutionary Multi-Criterion Optimization*, pp. 110–125, Cham, 2015. Springer International Publishing. ISBN 978-3-319-15892-1. doi: 10.1007/978-3-319-15892-1_8.

Justin K. Pugh, Lisa B. Soros, and Kenneth O. Stanley. Quality Diversity: A New Frontier for Evolutionary Computation. *Frontiers in Robotics and AI*, 3, 2016. ISSN 2296-9144. URL `https://www.frontiersin.org/articles/10.3389/frobt.2016.00040`.

Lennart Purucker and Joeran Beel. Assembled-OpenML: Creating Efficient Benchmarks for Ensembles in AutoML with OpenML, July 2023. URL `http://arxiv.org/abs/2307.00285`. arXiv:2307.00285 [cs].

Lennart Purucker, Lennart Schneider, Marie Anastacio, Joeran Beel, Bernd Bischl, and Holger Hoos. Q(D)O-ES: Population-based Quality (Diversity) Optimisation for Post Hoc Ensemble Selection in AutoML, August 2023. URL `http://arxiv.org/abs/2307.08364`. arXiv:2307.08364 [cs].

David Salinas and Nick Erickson. TabRepo: A Large Scale Repository of Tabular Model Evaluations and its AutoML Applications, November 2023. URL `http://arxiv.org/abs/2311.02971`. arXiv:2311.02971 [cs].

Yu Shen, Yupeng Lu, Yang Li, Yaofeng Tu, Wentao Zhang, and Bin CUI. DivBO: Diversity-aware CASH for Ensemble Learning. In S. Koyejo, S. Mohamed, A. Agarwal, D. Belgrave, K. Cho, and A. Oh (eds.), *Advances in Neural Information Processing Systems*, volume 35, pp. 2958–2971. Curran Associates, Inc., 2022. URL `https://proceedings.neurips.cc/paper_files/paper/2022/file/13b2f88be223cd2b4d6be67b56e02fa8-Paper-Conference.pdf`.

E. Zitzler and L. Thiele. Multiobjective evolutionary algorithms: a comparative case study and the strength Pareto approach. *IEEE Transactions on Evolutionary Computation*, 3(4):257–271, November 1999. ISSN 1089778X. doi: 10.1109/4235.797969. URL `http://ieeexplore.ieee.org/document/797969/`.

## A  Method

### A.1  Enhancements to GES

The original GES algorithm optimizes solely for predictive performance without considering other objectives, e.g. hardware costs. This can lead to ensembles that slightly improve accuracy but incur disproportionately higher inference times. To address this, we introduced two enhancements: generating a spectrum of solutions and extending GES to handle multiple objectives.

**Spectrum of Solutions**   GES usually outputs a single final ensemble. We modified it to record intermediate ensembles at each iteration, resulting in a set of $n$ ensembles when running for $n$ iterations. Each ensemble adds one model and achieves a higher validation score than the previous one. This creates a spectrum of solutions with varying trade-offs between predictive performance and hardware cost.

**Multi-GES**   To explicitly account for hardware efficiency, we extended GES into Multi-GES, introducing a weighted multi-objective scoring system that balances predictive performance and inference time. Both metrics are normalized to ensure scale comparability, and static weights $(\alpha, \beta)$ control their relative importance.

Algorithm 1 outlines the procedure, where red lines mark the multi-objective components and blue lines indicate the recording of intermediate ensembles.

## B  Experimental Setup

**Hypervolume (HV):**   HV measures the size of the objective space dominated by the solutions on the Pareto front, relative to a chosen reference point. It captures both convergence (how close solutions are to the optimum) and diversity (how well they cover the trade-off surface) in a single scalar value. A larger HV indicates that a method has found better trade-offs between objectives, making it one of the most widely used and Pareto-compliant indicators in multi-objective optimization.

---

**Algorithm 1** Multi-GES

---

1: Initialize an empty ensemble $E$.
2: Initialize the set of candidate models $M$.
3: Initialize an empty list of ensembles $\mathcal{E}$.
4: Set time weight $\beta \in [0, 1]$.
5: Set performance weight $\alpha = 1 - \beta$.
6: **while** not finished **do**
7:     Initialize $E'$ as a temporary ensemble.
8:     Initialize $J_{\text{best}} \leftarrow \infty$.
9:     **for** each model $m$ in $M$ **do**
10:         Form $E' = E \cup \{m\}$.
11:         Compute normalized performance $P_{E'}$.
12:         Compute normalized time $T_{E'}$.
13:         Compute objective value: $J_{E'} = \alpha \cdot P_{E'} + \beta \cdot T_{E'}$.
14:         **if** $J_{E'} < J_{\text{best}}$ **then**
15:             Update $J_{\text{best}} \leftarrow J_{E'}$.
16:             Update $E^* \leftarrow E'$.
17:         **end if**
18:     **end for**
19:     Set $E \leftarrow E^*$.
20:     Append a copy of $E$ to $\mathcal{E}$.
21: **end while**
22: **return** The set of ensembles $\mathcal{E}$.

---

**Inverted generational distance plus (IGD+):** IGD+ quantifies how closely the solutions produced by a method approximate a reference Pareto front by measuring the average distance from each point on the reference front to its nearest point in the approximation. It evaluates both how well a method converges to the optimal trade-offs and how evenly its solutions cover the front. Unlike the original IGD, IGD+ only penalizes inferior dimensions, ensuring consistency with Pareto dominance.

## C   Results

**Main Results**

Figure 15: Normalized improvement over the single-best baseline method.

