# OpenReview forum: "HAPEns: Hardware-Aware Post-Hoc Ensembling for Tabular Data"
_TMLR — Withdrawn by Authors_

### Review · Reviewer_prTN · 2025-11-27

**Summary Of Contributions:**

Summary:

The paper studies the problem of hardware-aware ensemble selection for tabular learning. They modify the search space of an existing evolutionary quality-diversity (QD) optimization algorithm (Fontaine et al. 2019) to be hardware-aware. As a result, HAPEns effectively enumerates the Pareto frontier compared to non-hardware-aware ensembling methods.

**Audience:**

Yes

**Audience Explanation:**

Please see S1-S3.

**Broader Impact Concerns:**

None, as far as I can check.

**Claims And Evidence:**

No

**Claims Explanation:**

Strengths:

S1. The problem is well-motivated, clearly defined, and has clear implications for several applications.

S2. The empirical evaluation is extensive, incorporating 83 tabular datasets, 1500+ models, and multiple metrics of Pareto frontier quality.

S3. The authors have ensured reproducibility through open-source code and data.

Opportunities for Improvement:

O1. Please clarify whether there are any characteristics of ensemble search in tabular learning that is challenging or different compared to other multi-objective optimization problems. Has HAPEns been designed to be tailored to tabular ensembling?

O2. HAPEns considers one hardware objective. Whereas, other objectives such as memory are quite important. The paper needs to show that HAPEns can effectively enumerate the Pareto frontier with respect to more than one hardware objectives at the same time.

O3. It is not clear why the paper chooses to frame the problem as primarily a QD problem, as opposed to a multi-objective optimization (MOO) problem. The paper needs to compare HAPEns with modern MOO methods, in terms of optimization time vs HV/IGD+.

O4. Please clarify why the 7x7 behavior grid is chosen, and not any other bin count in each direction. The paper needs to provide a sensitivity study with respect to the bin granularity.

Assessment:

While the problem is well-motivated and the overall experimental setup is sound, the authors need to further clarify design decisions and expand the experiments for their claim to be well-supported.

**Requested Changes:**

Please see O1-O4.

---

### Review · Reviewer_bXNe · 2025-12-07

**Summary Of Contributions:**

The paper focuses on post-hoc ensemble selection for tabular prediction when deployment is constrained by hardware cost.  The contributions are: (1) The authors study an interesting problem (2) they propose HAPEns, a method that searches over ensembles of pre-trained base models in a two-dimensional “behavior space” defined by average loss correlation and a hardware metric, using niches together with crossover and mutation over repetition vectors to approximate the Pareto front. (3) It provides an extensive empirical study on 83 tabular datasets showing that HAPEns achieves better accuracy–efficiency trade-offs than 4 baselines.

**Additional Comments:**

This is an interesting and practically relevant problem. However, the proposed solutions are quite naive, and the writing needs improvement. The paper also lacks formal analysis of the search procedure and of how well it approximates the Pareto front. My recommendation for acceptance depends on how well the authors address the issues listed in the “Requested Changes” section.

**Audience:**

Yes

**Audience Explanation:**

People who work on AutoML, ensembles, or resource-constrained deployment for tabular models will be interested. That said, hardware-aware ensembling for tabular data is a fairly narrow slice of the overall AutoML and ensemble directions; the impact would be higher if the ideas were shown to extend to more general task settings.

**Broader Impact Concerns:**

I don’t see major ethical or broader-impact concerns specific to this work.

**Claims And Evidence:**

Yes

**Claims Explanation:**

To some degree, I would say. The main claims are supported by experiments on a large number of tabular datasets, with fairly comprehensive experiment design (e.g., different hardware metrics, variants of the proposed method). However, the baselines are relatively simple: Single-Best, standard Greedy Ensemble Selection that does not itself optimize hardware constraints, and Multi-GES, which is a naïve static weighting of accuracy and cost. Including more advanced multi-objective or hardware-aware ensemble selection methods would strengthen the empirical evidence and better validate the advantages of the proposed approach. Overall, the current results are convincing for the settings considered, but the baseline choice limits how strong the empirical claims can be.

**Requested Changes:**

(1) The topic is interesting, but the intuition behind using crossover and mutation on the repetition vectors is not clearly explained, and the proposed solution looks rather naïve.
(2) The method lacks any theoretical analysis or guarantees on how well it approximates the hardware-aware Pareto front.
(3) Although the method is hardware-aware, the evolutionary search itself introduces additional search cost; the paper does not analyze this complexity or quantify the extra overhead, either experimentally or theoretically.
(4) The baselines for multi-objective optimization and ensemble selection are relatively simple (Single-Best, GES*, Multi-GES, QDO-ES), so it is unclear whether this relatively naïve solution would still be competitive against more advanced multi-objective or Pareto-based approaches.
(5) The layout and writing of the paper could be improved for readability and aesthetics. In particular, the introduction and method sections are not well motivated or clearly organized: just for example, how “post-hoc ensembling” methods improve the performance and whether post-hoc ensembling necessarily requires large ensembles are not clearly explained, and the intuition behind each component of the method is under-motivated. In addition, Figure 2 is somewhat confusing, and page 6 currently contains only Figures 4 and 5; it would be better to rebalance text vs. figures or rearrange them to avoid large empty areas.

---

### Review · Reviewer_uuRr · 2025-12-09

**Summary Of Contributions:**

The submission proposes a post-hoc ensembling framework specifically tailored for the tabular domain. The primary objective of this work is to achieve an optimal trade-off between predictive accuracy and hardware efficiency. To accomplish this, the paper introduces a search mechanism (HAPEns) to find effective ensemble combinations. The experimental results presented suggest that the proposed method is capable of outperforming standard baseline methods in the selected settings.

Strengths
1. The problem of balancing model performance with memory usage and hardware constraints is highly relevant for real-world tabular applications where resources are often limited.

Weaknesses
1. The writing quality presents a significant barrier to understanding. Key methodological details are described vaguely. For example, the specific mechanics of the crossover and mutation operators lack mathematical rigor in the main text.
2. The presentation of the experimental section lacks a logical flow. The relationship between the exploratory analysis of model types in Figure 5 and the final optimization results is not clearly established.
3. There is a disconnect between the paper's motivation regarding hardware constraints and its validation. The authors explicitly state that they abstract away from specific hardware configurations. This means the claims regarding real-world applicability are not sufficiently verified by measurements on actual limited hardware.

**Audience:**

Yes

**Audience Explanation:**

The development of ensembling strategies that effectively balance performance with hardware usage is a topic of significant importance. The findings that memory usage serves as a particularly effective proxy metric for general hardware costs would be relevant to practitioners.

**Broader Impact Concerns:**

I have no concern for this part.

**Claims And Evidence:**

No

**Claims Explanation:**

The current evidence is insufficient to support the claims for the following reasons:

The writing needs to be significantly improved to allow for proper verification. The authors mention using a "two-point crossover limited to nonzero entries" and mutation via "rejection sampling" but fail to provide clear equations or a formal algorithm block to define these operations precisely. Furthermore the text references specific contexts like "D244_F3_C1530_100" without explaining what these codes represent to readers unfamiliar with the specific versioning of TabRepo.

The descriptions of experiments are confused. In the experiments the relation between the inference time distribution shown in Figure 5 and the aggregated "Hardware Score" used in the main results is not explicitly detailed. It is unclear how the theoretical inference times in Figure 5 translate to the actual optimization objectives in the evolutionary search.

The real-world application scenarios are not verified. The authors rely on normalized proxy metrics and admit that their experiments "abstract away from specific hardware configurations". Since the method frames itself as a solution for "tight resource constraints"  it is critical to validate these ensembles in an environment with actual hard limits on memory or latency rather than just optimizing a multi-objective function on a server

**Requested Changes:**

I recommend the following adjustments to strengthen the work:

Rewrite the Methodology section: The descriptions regarding the HAPEns search process via crossover and mutation must be re-written for clarity. Please provide formal definitions or pseudocode explaining how the "two-point crossover" operates on the weight vectors and how the "rejection sampling" is implemented.

Clarify Terminology: The context regarding "D244_F3_C1530_100" must be clearly defined within the text so the experimental setup is self-contained.

Verify Real-world Scenarios: The real-world application scenarios should be verified. The authors should attempt to deploy the resulting ensembles on a device with actual hardware constraints (or a rigorous simulator) to demonstrate that the "Hardware Score" translates to practical benefits. Relying solely on the "abstraction" mentioned in Section 5 is insufficient.

Justify Hyperparameter Choices: A clear explanation or justification for dividing the behavior space into exactly "7 by 7 niches" should be provided. The authors should verify if the results are sensitive to this grid size.

---

### Note · Authors · 2025-12-18

**Comment:**

We thank the reviewers for their detailed and thoughtful feedback. After careful consideration, we have decided to withdraw the paper at this stage.

The reviews made it clear to us that the paper’s intent was not communicated with sufficient precision, leading to several recurring misconceptions about its scope and claims. In particular, our work was frequently interpreted as an optimization-focused hardware study. In contrast, we intended to present an application-oriented, hardware-aware post-hoc ensembling pipeline explicitly designed for tabular data, benchmarked using model repositories well-known in the field of tabular data and post-hoc ensembling.

We realize that we did not convey the following aspects clearly enough:
* All methods are evaluated using concrete hardware measurements on the same system and the same data; specific hardware configurations themselves are not the focus of the paper, but holding them fixed allows us to demonstrate that the proposed approach consistently outperforms existing post-hoc ensembling methods under identical conditions.
* The contribution is a pipeline-level extension of post-hoc ensemble selection, rather than a new optimizer or theoretical multi-objective method.
* For tabular data, greedy ensemble selection (GES) remains the dominant and highly competitive baseline, and no optimizer-based methods are used in practice, nor are typical optimizer-based methods competitive baselines in related academic work.
* There are currently no hardware-aware post-hoc ensemble selection baselines to compare against.
* The work is motivated by practical deployment constraints in tabular pipelines, not by advancing general-purpose multi-objective optimization methodology.

Given this gap between our intended contribution and how it was perceived, we believe the paper would benefit from a substantial reframing. We therefore plan to refine the manuscript to more clearly distinguish pipeline design and applied data science from optimization, to better motivate the tabular post-hoc ensembling setting, and to avoid overstating claims that could be interpreted as hardware deployment validation.

We appreciate the reviewers’ input and view it as valuable guidance for strengthening a future version of this work. We thank the reviewers for their valuable time and for helping us improve the manuscript.

**Withdrawal Confirmation:**

I have read and agree with the venue's withdrawal policy on behalf of myself and my co-authors.